# Pharmacokinetics and Dose Proportionality Study of a Novel Antiparkinsonian Agent, a 1*H*-1,2,4-Triazol-3-ylthio-conjugate of Prottremine

**DOI:** 10.3390/molecules29184498

**Published:** 2024-09-22

**Authors:** Daria S. Gorina, Anastasiya V. Lastovka, Artem D. Rogachev, Alexandra V. Podturkina, Alla V. Pavlova, Oleg V. Ardashov, Nikolai S. Li-Zhulanov, Tatyana G. Tolstikova, Konstantin P. Volcho, Nariman F. Salakhutdinov

**Affiliations:** 1N. N. Vorozhtsov Novosibirsk Institute of Organic Chemistry, Siberian Branch of the Russian Academy of Sciences, Lavrentiev Ave., 9, 630090 Novosibirsk, Russia; d.gorina@g.nsu.ru (D.S.G.); rogachev@nioch.nsc.ru (A.D.R.); podturkina@nioch.nsc.ru (A.V.P.); pavlova@nioch.nsc.ru (A.V.P.); ardashov@nioch.nsc.ru (O.V.A.); lizhulan@nioch.nsc.ru (N.S.L.-Z.); tg_tolstikova@mail.ru (T.G.T.); volcho@nioch.nsc.ru (K.P.V.); anvar@nioch.nsc.ru (N.F.S.); 2Department of Natural Sciences, Novosibirsk State University, Pirogova str., 2, 630090 Novosibirsk, Russia

**Keywords:** Parkinson’s disease, LC–MS/MS, validation, pharmacokinetic, dose proportionality, bioavailability, monoterpene

## Abstract

The novel antiparkinsonian agent PA-96 is the focus of our research. PA-96 supported the survival of cultured naïve dopamine neurons, alleviated motor deficits in MPTP and haloperidol-based mice models of Parkinson’s disease, and increased the density of tyrosine hydroxylase positive neurons and dopamine concentration in the midbrain of an MPTP-damaged brain. In this work, an HPLC–MS/MS method was developed and validated, and the pharmacokinetics of the agent was investigated in mice after a single or multiple oral administration (*p.o.*) and intravenous injection (*i.v.*) at various doses. The dose proportionality was also evaluated after a single *p.o.* administration of three ascending doses (1, 5, and 10 mg/kg) and a single *i.v.* injection of two doses (1 and 10 mg/kg); also, the bioavailability was estimated. The disproportionality of pharmacokinetic parameters could be explained by the saturation of active centres of enzymes or receptors binding the substance: at low doses, part of the compound is bound, leaving a small amount circulating in blood, and rapidly metabolised and/or bound too. The bioavailability of PA-96 was c.a. 7 and 35% for the doses of 5 and 10 mg/kg, correspondingly.

## 1. Introduction

Parkinson’s disease (PD) is one of the fastest growing neurological diseases in the world [1,2,3,4]. PD is characterised by movement disorders, such as tremors, stiffness, and difficulty with balance and coordination, all caused by the loss of dopaminergic neurons [5,6,7]. In addition, patients experience non-motor symptoms, including sleep disturbances, pain, fatigue, and psychiatric symptoms [8,9,10,11]. According to the Global Burden of Disease (GBD) report from 2019, over 8.5 million people worldwide suffer from PD [12]. The number of people with PD has increased 2.5 times over the past 30 years. With an aging population, this number is expected to double to 12.9 million by 2040 [13]. Moreover, other factors such as genetic mutations, endogenous factors, chemical toxicants, and viral Parkinsonism will only exacerbate this trend [14,15,16,17].

There is no convincing evidence to support the efficacy of drugs with neuroprotective, neurorestorative, and disease-modifying effects. For some groups of symptomatic drugs (type B monoamine oxidase inhibitors, dopamine receptor agonists, adamantane derivatives, dopa drugs and its derivatives), there is only indirect evidence that a neuroprotective potential exists [18,19,20,21]. In addition, these treatments often have serious side effects [22,23,24]. For instance, Levodopa is associated with significant side effects, including motor fluctuations and dyskinesia, arrhythmias, and psychotic and paranoid reactions [25,26,27]. By all means, the search for and development of new compounds with high efficacy and low toxicity remain challenges for medicinal chemistry.

Developing new compounds that have an influence on the central nervous system (CNS) is a more complex process in contrast to many other therapeutic areas. Preclinical and clinical trials are complicated by the necessity to prove that a compound can pass through the blood–brain barrier (BBB) as well as reach damaged dopamine neurons [28,29,30,31]. In addition, determining an effective dosing regimen and studying ADME (absorption, distribution, metabolism, and excretion) processes in vivo require obtaining pharmacokinetic parameters. Within the context of this research, it is crucial to develop and validate sensitive and selective bioanalytical methods according to standards in order to ensure accurate and precise results [32,33].

Earlier, our group synthesised (1*R*,2*R*,6*S*)-3-methyl-6-(prop-1-en-2-yl)cyclohex-3-ene-1,2-diol starting from (−)-verbenone. This agent demonstrated effective antiparkinsonian activity at a dose of 20 mg/kg comparable to that of L-DOPA administered at doses of 50–100 mg/kg in various mouse and rat models, such as MPTP-, haloperidol-, and rotenone-induced models [34,35]. The agent, named Prottremine, has low acute toxicity (LD_50_ of 4250 mg/kg in mice) [36], and based on its properties, we have conducted its preclinical studies. Moreover, to date, Prottremine has successfully passed the first phase of clinical trials [37].

Investigations of Prottremine analogues revealed that the (1*R*,2*R*,6*S*) configuration significantly influences its pharmacological activity [34]. In addition, several Prottremine derivatives were synthesised, which also showed antiparkinsonian activity [38]. Recently, a Prottremine derivative with a 1*H*-1,2,4-triazol-3-ylthio substituent at the allyl position at C-2 (agent PA-96, Figure 1) was synthesised using an original three-step stereoselective procedure [39]. The product, PA-96, proved to be more active than Prottremine and displayed this effect at a dose as low as 1 mg/kg. It showed improved potency in an in vitro survival assay for dopamine neurons and in animal MPTP and haloperidol models of PD, and it readily crossed the BBB (K_b,brain_ was equal to 0.119 ± 0.010). Nevertheless, only a preliminary pharmacokinetic study was performed in [39], and it was necessary to develop a dosage regimen and determine the key pharmacokinetic parameters for the compound. The main objectives of this study were to validate an LC–MS/MS method for quantification of PA-96 in mouse blood as well as to investigate the pharmacokinetics, dose proportionality, and bioavailability of the agent.

## 2. Results and Discussions

### 2.1. Mass Spectrometric and Chromatographic Conditions

The positive ESI mode was selected for sample analysis. Firstly, a PA-96 solution was infused directly into the mass spectrometer; the mass spectrum was obtained in the +Q1 mode (without fragmentation), and the formation of two molecular ions with *m*/*z* = 252.2 and 274.1 was observed, corresponding to the formation of the protonated form [M+H]^+^ and to the adduct [M+Na]^+^, respectively (Appendix A). During the fragmentation of the molecular ion with *m*/*z* = 252.2 at low, medium, and high energy levels, the formation of fragment ions having *m*/*z* = 234.1, 151.2, 133.1, and 102.1 was observed (Appendix A). When the molecular ion with *m*/*z* = 274.1 was fragmented, intense product ions with *m*/*z* = 256.3, 124.1, and 105.1 were obtained (Appendix A).

At the first stage of optimising the chromatographic conditions, an analysis was performed with detection of all previously selected MRM transitions, including fragmentation of both molecular ions observed for PA-96. It was noted that the transitions based on the fragmentation of the molecular ion with *m*/*z* = 274.1 had significantly lower intensity. When examining them separately, it became clear that the signal in the chromatogram was very weak and contained noise signals, possibly belonging to impurities. Therefore, for the following analyses, all MRM transitions of the molecular ion 274.1 were excluded from the method. The PA-96 MRM transition with *m*/*z* 252.2→102.1 had the highest intensity, so it was selected as a main quantitative MRM transition. Transitions with *m*/*z* 252.2→133.1/151.2 were chosen as qualifiers (Figure 2). The parameters for the detection of the IS in MRM mode were optimised earlier [40].

The chromatographic conditions were optimised to operate acceptable peaks with maximum intensity and reasonable elution time. PA-96 has good solubility in inorganic solvents such as methanol and acetonitrile. However, using acetonitrile was less preferable because of its tendency of causing a breakthrough of the internal standard (2-Ad) as observed earlier [41]. Thus, methanol was chosen as the organic component for the mobile phase. Various mobile phase compositions and elution conditions have been tested, i.e., different volume percentage of water and methanol, with or without formic acid, isocratic or gradient elution. Formic acid with a concentration of 0.1% in both mobile phase solvents is required to increase ionisation in the positive ESI mode and improve signal response. Gradient elution conditions were established for the final analysis using mobile phases consisting of two solvents: methanol with 0.1% formic acid (solvent B) and water with 0.1% formic acid (solvent A), starting with a 90:10 (*v*/*v*) ratio. The flow rate was set to 200 μL/min. Under these conditions, PA-96 eluted at 8.7 min with a total analysis runtime of 12 min including the column equilibration step. The injection volume was 1 μL.

### 2.2. Sample Preparation Method and Biostability Study

There are many methods for whole blood sample preparation, and the choice mainly depends on the type and volume of the sample, the physicochemical properties of the analytes, and the analytical platform used for the analysis. In our experiment, samples were prepared using dried blood spots (DBS), a common method in preclinical studies of new bioactive agents in small animals. Also, a simple protein precipitation (PPT) method using methanol and zinc sulfate (ZnSO_4_) was used in our work as an alternative approach for whole blood sample preparation. In order to extract PA-96 from DBS samples, methanol, acetonitrile, methanol with 0.1% (*v*/*v*) formic acid, and water with the following concentration, as well as methanol with 0.1% (*v*/*v*) formic acid with the following concentration were used. The peak area of PA-96 in the chromatograms of the samples obtained after PPT significantly exceeded the peak area after all considered methods of sample preparation (Figure 3).

A crucial factor to consider is the stability of the compound in biological matrices. Both whole blood and plasma may contain enzymes able to metabolise the substance, and the metabolism rate can be relatively high [42]. This can result in a rapid decrease in the substance concentration and an incorrect calibration curve. Additionally, if blood samples from experimental animals are dried as spots, metabolism during drying may lead to incorrect analyte concentrations [43]. Considering these factors, we investigated the stability of the agent PA-96 in whole mouse blood. During the stability investigation, the PA-96 concentration in blood was consistent within the accuracy of analysis error and obtained values rounding (Figure 4).

### 2.3. Method Validation

The development and validation of the PA-96 determination method using HPLC–MS/MS along with its application in pharmacokinetic research are undoubtedly critical steps in studying the compound’s bioavailability and dose proportionality. In preclinical research involving CNS-targeting compounds, proving blood–brain barrier (BBB) penetration is essential to demonstrate targeted action. Previously, we developed HPLC and MS conditions for PA-96 determination and obtained its pharmacokinetics in whole mouse blood and brain. We also proved the BBB penetration of the new molecule PA-96. *K_b,brain_* was calculated as a ratio of the area under the brain/blood pharmacokinetic curves from 0 to 24 h (*K_b,brain_* = AUC*_brain_*/AUC*_blood_*). *K_b,brain_* was equal to 0.119 ± 0.010 (*p* = 0.95) [39]. In the present work, we validated the PA-96 determination method in whole blood for the following parameters: selectivity and specificity, linearity, lower limit of detection and lower limit of quantification, extraction recovery, matrix effect, precision and accuracy, carryover effect, and stability. The validated method was used to obtain the PA-96 pharmacokinetics at single and multiple, and oral and intravenous administration at various doses, enabling the investigation of dose proportionality and calculation of bioavailability.

#### 2.3.1. Selectivity

The selectivity of the developed method was investigated by the analysis of six blank blood samples obtained from different animals in the LC–MS/MS conditions mentioned above. The chromatogram obtained in this analysis contained two peaks with MRM transitions corresponding to the agent PA-96 (Appendix A). The presence of these peaks apparently indicates the extraction of two native metabolites having the same MRM transitions as PA-96. Luckily, the impurity with the most intense MRM transition of PA-96 (252.2 → 102.1) was eluted at approximately 8.1 min, while the retention time of PA-96 was approximately 8.8 min, and the peaks were separated to the baseline (Appendix A). The second impurity that eluted close to PA-96 (retention time c.a. 8.6 min) did not have this MRM transition and, therefore, did not contribute to the quantifier signal of the analyte (Appendix A). No signals that would interfere with that of the 2-Ad at 6.0 min were observed in the obtained chromatograms (Appendix A). As such, we were able to suppose that the developed method was valid and met the selectivity criterion.

#### 2.3.2. Calibration Curve

In order to obtain the calibration curve, the spikes of PA-96 in whole murine blood containing the substance in the concentrations of 1, 5, 10, 25, 50, 100, 250, 500, 750, 1000, 1500, and 2000 ng/mL were prepared. The samples were processed according to the developed protocol and analysed.

When a linear calibration was achieved for all calibrators, it was found that the accuracy bias for those with concentrations of 1, 5, and 10 ng/mL exceeded the allowed level of 20%. Therefore, the 25 ng/mL concentration was chosen as the LLOQ, with the signal/noise ratio being 190. The calibration curve is described by the equation *y* = 1.5 × 10^−4^*x* + 1.5 × 10^−3^, and the correlation coefficient was 0.9975 (Appendix A).

#### 2.3.3. Recovery and Matrix Effect

The extraction recovery and matrix effect of the PA-96 from whole blood were assessed for the LLOQ and HQC samples. The obtained recovery was measured to be from 49 to 57% at LLOQ concentrations (25 ng/mL) and 68.3–70.7% at high concentrations (1500 ng/mL). The CV calculated for recovery was not higher than 8% at the LLOQ and 1.7% at the HQC (acceptance value is ≤20% and 15%, respectively). The obtained mean value of the matrix factor was 0.70 ± 0.03 at the LLOQ (25 ng/mL) and 0.448 ± 0.013 at the HQC (1500 ng/mL). The CV of the matrix effect did not exceed 4% at the LLOQ and 2.9% at the HQC. The obtained results are summarised in Appendix A.

#### 2.3.4. Accuracy and Precision

Accuracy and precision were determined by analysing the quality control (QC) samples and back calculation of PA-96 concentrations using the obtained calibration curve. Intra-day results indicated that the relative error values were below 12% for blood samples and the deviation values were below 15% at various concentration levels. Inter-day results showed that the percentage relative error values were less than 14% and the deviation values were less than 7% at different concentration levels (Appendix A).

#### 2.3.5. Carryover

In all matrices, the PA-96 peak detected in the blank sample injected directly after the highly concentrated PA-96 showed less than 20% of the peak area produced at its LLOQ. These results suggested that the carryover effect is negligible for analysing PA-96 and IS, reflecting the reliability of the separation column and analysis method.

#### 2.3.6. Stability of Prepared Samples

The stability of PA-96 in the prepared samples was investigated at different storage conditions, as mentioned in the experimental methods section. It was discovered that the PA-96 concentration did not change more than 11% within 7 days in all conditions. Since the CV did not exceed the acceptance criterion of 15%, we concluded that the PA-96 solutions after the sample preparation were stable (see Figure 5).

### 2.4. Investigation of Pharmacokinetics of PA-96

After the development and validation of the method for determination of PA-96 in mice blood, a preliminary pharmacokinetic study was carried out. The agent PA-96 was administered to two mice *per os* as a suspension in Tween-80 at a dose of 10 mg/kg, and blood samples were taken from the animals at 0.5, 1, 1.5, 2, 2.5, 3, 3.5, 4, 5, 5.5, and 24 h after administration. Four 10 µL samples were taken at each timepoint: two were spotted onto the paper and dried, and two others were precipitated using the ZnSO_4_ solution. All prepared samples were analysed, and the graph representing the PA-96 peak area values was plotted (Figure 6). As can be observed in Figure 6, the maximum concentration of the compound was observed at the first timepoint (30 min) after administration. Moreover, the PA-96 peak area in the chromatograms of the DBS samples (Figure 6, red line) was significantly lower than that measured for the samples after PPT with methanol and zinc sulphate (Figure 6, blue line). As such, the PPT method for whole blood samples was chosen for the next experiments.

The obtained data demonstrate a rapid absorption of PA-96 in mice and complete elimination after 2.5 h. Based on these results, we hypothesised that the observed timepoint for maximum concentration may have not been determined correctly; therefore, we decided to increase the number of timepoints of blood sampling during the first hour after substance administration. In the following experiment, the agent PA-96 was administered *p.o.* to three groups of animals at the doses of 1 mg/kg (*n* = 2), 5 mg/kg (*n* = 6), and 10 mg/kg (*n* = 4), respectively. After administration, blood samples were taken from the tail vein of each animal at 5, 10, 20, 30, 40, 50, 60, 75, 90, 120, and 150 min, and the samples were prepared following the PPT protocol.

Analysis of blood samples taken from the animals that received the agent PA-96 at a dose of 1 mg/kg showed a signal lower than the LLOQ. This indicates rapid absorption, distribution, and, probably, metabolism of the compound, as the concentration of its part circulating in blood was below the LOD of the method. When the agent PA-96 was administered at doses of 5 mg/kg and 10 mg/kg, its maximum concentrations were observed 5–10 min after administration (Figure 7). This indicates an extremely rapid absorption of the substance in mice, which may provide its quick activity. The maximum concentration of the compound administered at a dose of 5 mg/kg was approximately 36 ng/mL, and that observed after administration of a dose of 10 mg/kg was approximately 588 ng/mL. These results indicate a consistent, disproportional relationship between the tested doses of PA-96 (5 and 10 mg/kg).

The pharmacokinetic parameters calculated for the compound are represented in Table 1. As can be observed from the data, a twofold increase in the dose resulted in an increase in both C_max_ and AUC by an order of magnitude. PA-96 rapidly enters the bloodstream, and its complete elimination at oral administration is observed in 2.5–3 h. At small dosages (5 mg/kg), binding to active centres into complexes occurs; therefore, there are fewer free molecules in the bloodstream. This promotes long-term removal of the drug from the bloodstream (t_1/2_ = 44 min and Cl/F__obs_ = 3.3 × 10^−3^). At high dosages (10 mg/kg), most of the active centres are bound, but some part of PA-96 is free and capable of filtering and faster excretion (t_1/2_ = 15 min and Cl/F__obs_ = 6.4 × 10^−4^).

As the agent PA-96 showed both rapid absorption and excretion, it became obvious that, in order to use this compound as a potential drug, it was necessary to attain a more gradual admission of the compound, which requires the development of a prolonged formulation or multiple dose administration. As an alternative, multiple administration of the agent at lower doses can be tested.

In our work, an experiment with double administration of the agent PA-96 was carried out. The substance was administered to mice (*n* = 6) two times at doses of 5 mg/kg within the interval of 4 h as a formulation in Tween-80. After the first administration, blood was collected from the tail vein of each animal at 5, 10, 20, 30, 40, 50, 60, 75, and 90 min, and 2, 2.5, 3, 3.5, and 4 h. After the administration of the second dose, blood was collected at the same timepoints, and all samples were prepared following the PPT protocol.

Figure 8 shows the concentration–time profile of PA-96 in mice after repeated administration of the agent as described above. It can be observed that each injection of the agent results in a profile similar to each other and, simultaneously, to the profile shown in Figure 7 for the same dose. This means that the agent is fully eliminated in 4 h after administration at this dose and no residual amount is left in the mice, which could influence the second dose.

The pharmacokinetic parameters of PA-96 were calculated for two regions of the curve separately and are presented in Table 2. The results indicate that the values obtained are close to each other and to those obtained in the experiment with a single administration of the compound at a dose of 5 mg/kg (Figure 7 and Table 1). Notably, double administration of the agent PA-96 did not result in a proportionate increase in both its C_max_ and, more importantly, AUC_0-∞,_ indicating an absence of its accumulation.

The disproportionality of the pharmacokinetic parameters can be explained by the saturation of the active centres of the enzymes or receptors binding the substance: at low doses, part of the compound is bound, leaving a small amount circulating in blood, and rapidly metabolised and/or bound too. When the dose is increased, the fraction of the bound agent becomes smaller after saturation of the binding centres, and the fraction of the free compound circulating in the bloodstream increases. As a result, the observed C_max_ can be an order of magnitude higher. This allows for us to hypothesise that the two injections can be considered as independent, indicating complete distribution and/or metabolism and elimination of the agent 4 h after its administration to mice.

To evaluate the bioavailability of the agent PA-96, a suspension of the compound with Tween-80 was administered intravenously at doses of 1 and 10 mg/kg. After administration, blood in an amount of approximately 10 µL was collected from the tail vein of each animal at 5, 10, 15, 30, 45, 60, 90, 120, and 150 min, and samples were prepared following the PPT protocol.

Analyses of the obtained samples showed that blood taken from the animals receiving 10 mg/kg PA-96 contained the compound in concentrations exceeding the validated calibration range. Therefore, it was necessary to widen the operating concentration range of the calibration curve. To quantify PA-96 in these blood samples, we prepared and analysed spikes containing the agent in concentrations of 530, 1325, 2650, 3530, 8800, 10,600, and 21,200 ng/mL, which were used to construct a high-concentration calibration curve. In addition to these calibrators, quality control samples were prepared containing PA-96 at the concentrations of 1760, 5300, and 17,100 ng/mL (QCL, QCM, and QCH, respectively). The calibration curve was built over all calibrators using a quadratic approximation (Appendix A), and intra-day accuracy and precision were found to be appropriate (Appendix A). As the objective of building this calibration curve was to evaluate the concentration of PA-96 only in several samples, the full validation of this expansion of the developed bioanalytical method was not performed.

Figure 9A,B show concentration–time profiles for PA-96 after its *i.v.* injection to mice at the doses of 1 mg/kg and 10 mg/kg, respectively. As can be observed in the graphs, the substance is absorbed rapidly, and its concentration decreases twofold within 10–15 min after injection. Afterwards, the concentration of PA-96 decreases smoothly and reaches the level of several nanograms per mL 1 h after administration.

The pharmacokinetic parameters calculated for PA-96 administered intravenously are presented in Table 3. As shown by the data, when the dose of the substance was increased tenfold, the obtained values for C_max_ and AUC also increased disproportionately, namely, C_max_ increased by twentyfold, and the AUC value increased by approximately 25 times. The pharmacokinetic parameters of the agent PA-96 obtained in all experiments are summarised in Appendix A.

Overall, *i.v.* injection led to a significantly higher blood concentration than the *p.o.* route, as reflected by higher C_max_ values of more than 7700 ng/mL in comparison with c.a. 588 ng/mL (the values are for the dose of 10 mg/kg). In both administration routes, the pharmacokinetic profile of PA-96 indicates a rapid absorption phase, with the maximum peak concentration in the blood (T_max_) observed several minutes after both *i.v.* and *p.o.* administration. The obtained results showed that the PA-96 concentration increased disproportionally when the agent was administered intravenously.

The bioavailability of PA-96 administered *p.o.* was assessed via the AUC value obtained for *i.v.* injection of the compound at a dose of 1 mg/kg. The calculated values were c.a. 7% and 35% for the doses of 5 mg/kg and 10 mg/kg, correspondingly, confirming the non-linear behaviour of the compound in the dose–proportionality experiments.

## 3. Materials and Methods

### 3.1. Materials and Reagents

2-Adamantylamine hydrochloride (2-Ad), used as an internal standard (IS), was purchased from Sigma-Aldrich (Burlington, VT, USA). Analytical grade methanol was purchased from Merck (Darmstadt, Germany), and zero-grade acetonitrile was purchased from Cryochrom (Saint-Petersburg, Russia). Formic acid was purchased from Panreac (Barcelona, Spain,). Zinc sulphate was purchased from Sigma-Aldrich (Burlington, MA, USA). High-purity water was prepared using a Direct-Q 3 UV system (Millipore S. A. S., Molsheim, France). Whatman Protein Saver 903 Cards were purchased from Sigma Aldrich and used for dried blood spot sampling. PA-96 was synthesised according to the previously published procedure [39].

### 3.2. Instrumentation and LC–MS/MS Conditions

The LC–MS/MS analysis was carried out using a Shimadzu LC-20AD Prominence chromatograph equipped with a cooling autosampler, binary gradient pump, and column oven (Shimadzu, Japan) coupled with a 3200 QTRAP mass spectrometer (AB SCIEX, Framingham, MA, USA) equipped with an electrospray ionisation source (ESI). The mass spectrometer worked in positive mode and multiple reaction monitoring (MRM) detection mode for quantitation. Analyst 1.6.2 software (AB SCIEX, Framingham, MA, USA) was used for instrument control and data acquisition, and MultiQuant 2.1 software (AB SCIEX, Framingham, MA, USA) was used for quantitation. To calculate the main pharmacokinetic parameters, the Microsoft Excel plugin PKSolver was used [44].

Separation was achieved on a ProntoSil-120-5-C18 AQ column (2.0 × 75 mm, 5 μm particles, EcoNova, Novosibirsk, Russia) thermostated at +35 °C. The mobile phase was composed of water containing 0.1% formic acid (solvent A) and methanol containing 0.1% formic acid (solvent B). The gradient elution program was as follows: 0 min—10% (B); 0.5 min—10% (B); 7 min—98% (B); 9 min—98% (B); the flow rate was 200 μL/min. The samples were injected using an autosampler thermostated at +10 °C; the injection volume was 1 μL.

MS/MS detection was performed in positive ESI mode. The following parameters of ion source were set: ion source voltage was 5500 V, and the source temperature was +350 °C. For the agent PA-96 and IS, the optimised source parameters, viz. CUR (curtain gas), GS1 (Gas 1), and GS2 (Gas 2), were set at 20 psi. The CAD (collision-activated dissociation) gas was nitrogen, and the parameter was set as high. The quantitative analysis was carried out using the MRM mode of transitions from protonated precursor to product ions. MRM parameters were optimised for each mass transition, including collision energy (CE), dwell time, declustering potentials (DP), and collision cell exit potentials (CXP), which are summarised in Table 4. The dwell time was 100 ms for all transitions.

### 3.3. Working Solutions and Calibrators

The stock solution of PA-96 (1.0 mg/mL) was prepared by dissolving an accurately weighed amount of the substance in 100% methanol. To obtain the first set of working solutions with concentrations of 0.01, 0.05, 0.1, 0.25, 0.5, 1, 2.5, 5, 7.5, 10, 15, and 20 µg/mL, the stock solution of PA-96 was diluted with methanol. The second set of working solutions with concentrations of 0.009, 0.02, 1.3, 2.65, 5.3, 18, 27, 53, 106, and 212 µg/mL was also prepared, diluting the stock solution with methanol. To obtain quality control (QC) samples, working solutions with concentrations of 0.25, 0.75, 7.5, and 15 µg/mL were prepared. Other solutions with various concentrations for validation were prepared in the same manner. Calibration standards (calibrators) and QC samples were prepared by mixing 10 μL of each individual working solution and 90 μL of mice whole blood. Samples were stirred gently on an orbital shaker at 1000 rpm and left at 4 °C for 1 h for equilibration. The stock solution of IS with a concentration of 1.0 mg/mL was prepared by dissolving a weighed amount of 2-adamantylamine hydrochloride in pure methanol. The solution used for the protein precipitation procedure was a mixture of 2 parts of 0.2 M aqueous solutions of zinc sulphate per 8 parts of a solution of IS in methanol with a concentration of 10 µg/mL. All solutions were stored at −18 °C and brought to their ambient temperature before use.

### 3.4. Sample Preparation. Blood Preparation Protocol

#### 3.4.1. Protein Precipitation (PPT)

An aliquot of 10 µL of whole blood sample was added into an Eppendorf microtube (1.5 mL) containing 50 µL of the precipitation solution with IS. The sample was vortexed for 20–30 s (Vortex V-1 Plus, Biosan, Riga, Latvia) and then centrifuged for 10 min at 13,400 rpm (Eppendorf MiniSpin, Eppendorf, Germany). The supernatant was transferred into an insert of chromatographic vial and analysed.

#### 3.4.2. Dried Blood Spots (DBS) Preparation and Processing

An aliquot of 10 μL of a spiked sample was spotted onto a DBS card and allowed to dry at ambient temperature for at least 3 h. The whole blood spot was cut out and placed into an Eppendorf microtube (1.5 mL). Further, 200 μL of the solution of IS in ACN or MeOH or 0.1% HCOOH in MeOH was added to each tube. The disk with the solution of IS was shaken at 25 °C at 1400 rpm for 30 min. Then, it was centrifuged at 13,400 rpm for 10 min, and the extract was transferred into an insert of chromatographic vial and analysed. As an alternative to the described sample preparation protocol, 100 μL of deionised water or 400 μL of 0.1% HCOOH in MeOH was added to the whole blood spot in the tube. The disk with solution was shaken at 25 °C at 1400 rpm for 30 min. Then, it was centrifuged at 13,400 rpm for 10 min, and the obtained extract was transferred into another clean tube (1.5 mL) and evaporated to dryness in a vacuum. The residue containing analyte was dissolved in 100 μL of 50% aqueous methanol with IS, shaken for 30 min, and centrifuged. The supernatant was transferred into an insert of chromatographic vial and analysed.

### 3.5. Validation

The following method validation parameters were evaluated: selectivity and specificity, linearity, lower limit of detection and lower limit of quantification, extraction recovery, matrix effect, precision and accuracy, carryover effect, and stability [32].

#### 3.5.1. Selectivity and Specificity

In order to prove the method’s selectivity, the blank and zero calibrator were compared with the LLOQ at the retention times of the analyte and IS. The analyte signal in these samples (*n* = 6) should not be higher than 20% of the LLOQ. The response at the retention time of the IS in the blank should not be higher than 5% of the average IS response. Also, the IS signal should not fluctuate more than 50% of the average IS response.

#### 3.5.2. Linearity

A blank (no analyte, no IS), a zero calibrator (blank plus IS), and at least six non-zero calibrators covering the quantitation range, including the LLOQ, in every run were prepared. The concentration–response relationship should be fit by the most suitable and simplest regression model. The main acceptance criteria were the following: a total of 75% and a minimum of six non-zero calibrators should be ±15% of the nominal concentrations, but the LLOQ calibrator should be ±20% of the nominal concentrations; S/N (signal to noise) for the LLOQ ˃ 5 than for blank samples; *r* (correlation coefficient) > 0.990.

#### 3.5.3. Limit of Detection (LOD) and Lower Limit of Quantification (LLOQ)

LOD is the lowest analyte concentration that can be identified using the validated method. LLOQ is the lowest compound concentration that can be determined with satisfactory accuracy and precision. The LOD should yield a signal at least three times that of the blank, whereas the LLOQ should provide a signal at least five times that of the blank.

The accuracy should be ±20% of the nominal concentration (replicates ≥ 5). The precision should not be higher than 20% (replicates ≥ 5).

#### 3.5.4. Extraction Recovery

The extraction recovery was determined as the ratio of the analyte signal after sample preparation to the blank signal, which was spiked with the analyte post-extraction. The evaluation was carried out at the 3 concentration levels in at least 3 samples in each interval. The acceptance criterion was a CV ≤ 15% for each level.

#### 3.5.5. Matrix Effect (ME)

The matrix effect was determined as the ratio of the analyte signal with the matrix influence versus the analyte signal without matrix influence. The calculation was carried out at the HQC and LQC using 6 samples at each level.

#### 3.5.6. Precision and Accuracy

The evaluation of both accuracy and precision was performed by the following procedure: the calibration range was divided into 3 intervals (low ~25%, medium ~50%, and high ~75%); at least 4 QCs were prepared at each interval and measured at least 5 times; the accuracy and precision were evaluated inside and between each run at least 3 times. The main acceptance criterion was ± 15% of the nominal concentrations, except ±20% at the LLOQ (within run and between runs).

#### 3.5.7. Carryover

The possibility of carryover was checked by comparing the chromatogram of the blank, analysed after the non-zero calibrator with the maximum concentration, to the LLOQ. The acceptable signal on the blank chromatogram should not exceed 20% of the LLOQ.

#### 3.5.8. Stability of PA-96 in Mice Whole Blood and in Prepared Samples

To study the influence of the enzymes on the PA-96 stability in whole blood, mice blood spikes containing PA-96 in the concentration of 50 ng/mL were prepared. Aliquots of the spike were precipitated according to the protocol described above. Sampling was carried out at the following timepoints: 5, 10, 20, 30, 40, 50, 120, 180, and 240 min. The long-term stability of the samples was studied by the comparison of freshly prepared samples and samples prepared and kept in different conditions for 7 days. The storage conditions were the following: room temperature (+23 °C), household fridge (+4 °C), and Kelvinator (−70 °C). The acceptance criterion was a CV ≤ 15% for every day and any storing condition.

### 3.6. Animals and Pharmacokinetic Studies

CD-1 mice (male and female) were obtained from the SPF vivarium of the Institute of Cytology and Genetics, Russia. The mice weighed 25–30 g at the beginning of the experiment.

All animal experiments were carried out in accordance with the guidelines laid down in the legislation of the Russian Federation and European Communities Council Directive of 24 November 1986 (86/609/EEC) and were approved by the County Administrative Board of Southern Finland (license number: KEK15-022). The experiment was approved by the Local Bioethical Committee of the N.N. Vorozhtsov Novosibirsk Institute of Organic Chemistry SB RAS with the resolution number P-14-2023-01-02.

An accurately weighed amount of PA-96 was mechanically mixed with Tween-80 using a mortar and pestle followed by suspending the obtained sample in physiological solution. The animals (*n* = 2) received a single dose of 10 mg/kg *per os* (*p.o.*). The timepoints were 0.5, 1, 1.5, 2, 2.5, 3, 3.5, 4, 5, 5.5, and 24 h. At each timepoint, 10 µL of blood was taken from the tail vein, and the sample was prepared by DBS and PPT with ZnSO_4_ in methanol. Afterwards, the samples were analysed in accordance with the HPLC–MS/MS protocol.

In the next experiment, the sampling scheme was optimised by changing the timepoints as follows: 5, 10, 20, 30, 40, 50, 60, 75, and 90 min, and 2, 2.5, 3, 3.5, 4, and 5 h; this was in order to investigate the pharmacokinetics more thoroughly. The experiment design was not changed, but the sample preparation was carried out by the PPT method exclusively.

To investigate the pharmacokinetics of PA-96 in a multiple dosing regimen, the dosage form was prepared as described above, but only half was administered to the animals (*n* = 6). The first portion, corresponding to a dose of 5 mg/kg, was administered to the mice followed by blood sampling at 5, 10, 20, 30, 40, 50, 60, 75, and 90 min, and 2, 2.5, 3, 3.5, and 4 h after administration. At the latter timepoint, the second dose was administered, and the blood was sampled at the same timepoints as listed above.

In order to investigate the bioavailability and dose proportionality, PA-96 was administered *p.o.* at the dosages of 1, 5, and 10 mg/kg and intravenously (*i.v.*) at the dosages of 1 and 10 mg/kg. For each experiment, we used 6–7 mice. The blood was drawn from the tail vein at the following timepoints: 3, 5, 10, 15, 30, 45, 60, 90, 120, and 150 min. The samples were prepared and analysed.

## 4. Conclusions

A method for the quantification of the agent PA-96 in mice blood was validated and used for the pharmacokinetic study of the compound. Whole blood precipitation with a mixture of methanol and aqueous ZnSO_4_ was utilised for preparation of samples of 10 μL volume. The developed method was validated in terms of selectivity, calibration range, intra- and inter-day accuracy and precision, extraction recovery, and matrix factor. The pharmacokinetics of the agent was studied after the administration of the compound at doses of 1, 5, and 10 mg/kg *p.o.* as well as 1 and 10 mg/kg *i.v.*, and the bioavailability of PA-96 was evaluated to be c.a. 7% and 35% for the doses of 5 mg/kg and 10 mg/kg, correspondingly. The pharmacokinetics of the agent was found to be non-linear due to a disproportional increase in the AUC in experiments with increasing doses.

## Figures and Tables

**Figure 1 molecules-29-04498-f001:**
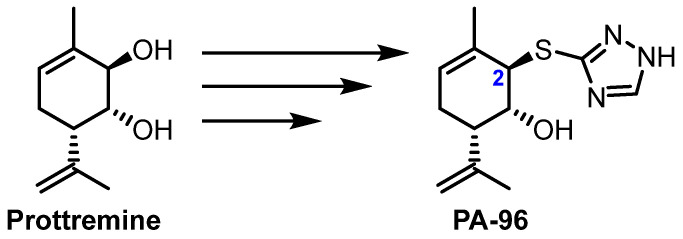
Prottremine and PA-96 structures.

**Figure 2 molecules-29-04498-f002:**
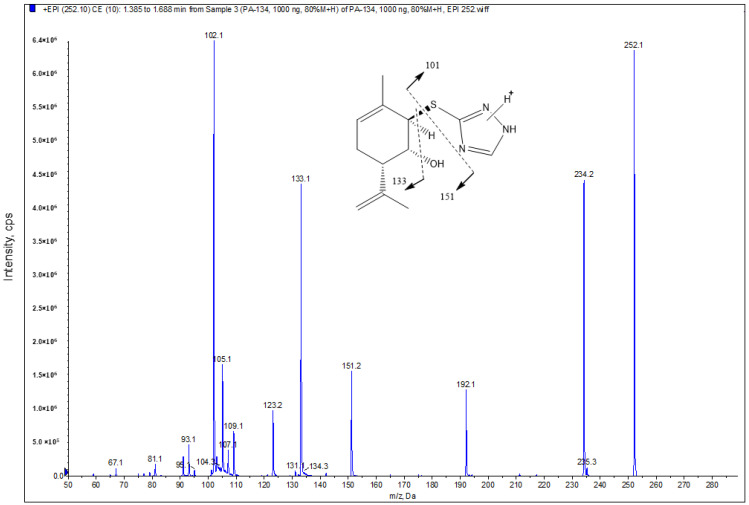
Mass spectrum and fragmentation scheme of the compound PA-96.

**Figure 3 molecules-29-04498-f003:**
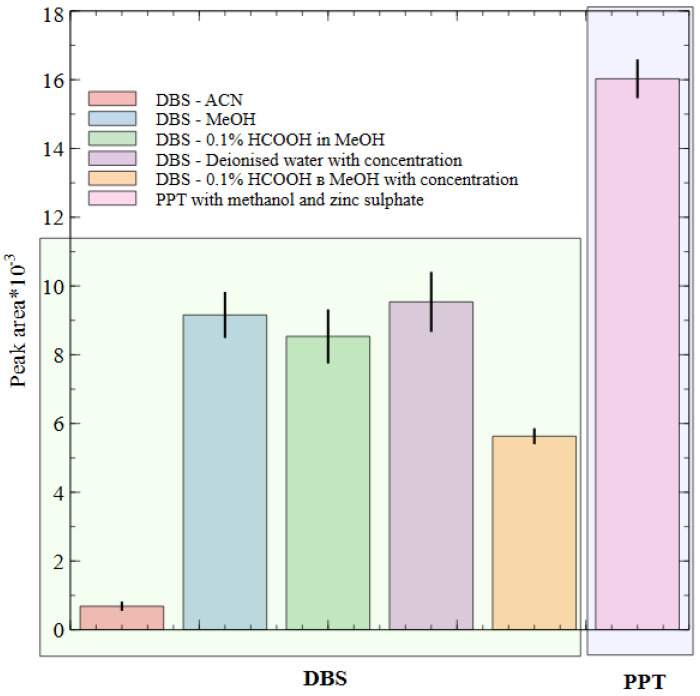
Peak area of PA-96 after different sample preparation methods.

**Figure 4 molecules-29-04498-f004:**
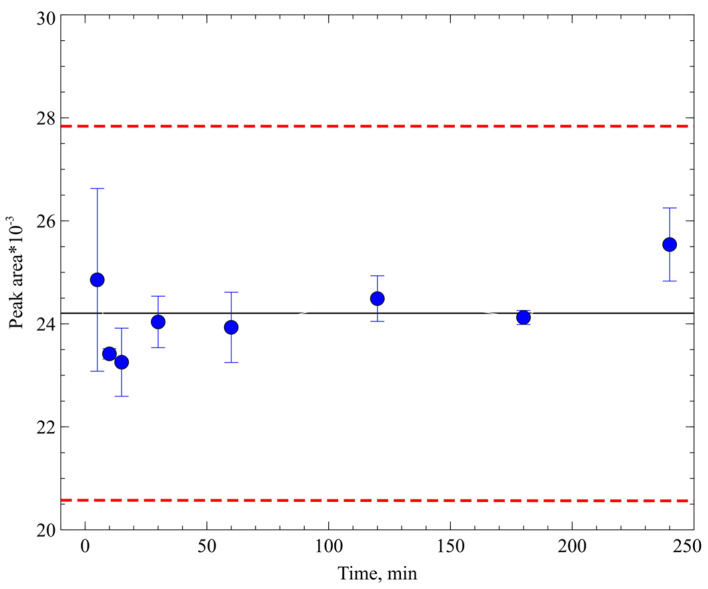
Peak area values of PA-96 in chromatograms of blood samples after spike preparation. Black line is the mean value, and red dotted lines indicate mean ± 15%.

**Figure 5 molecules-29-04498-f005:**
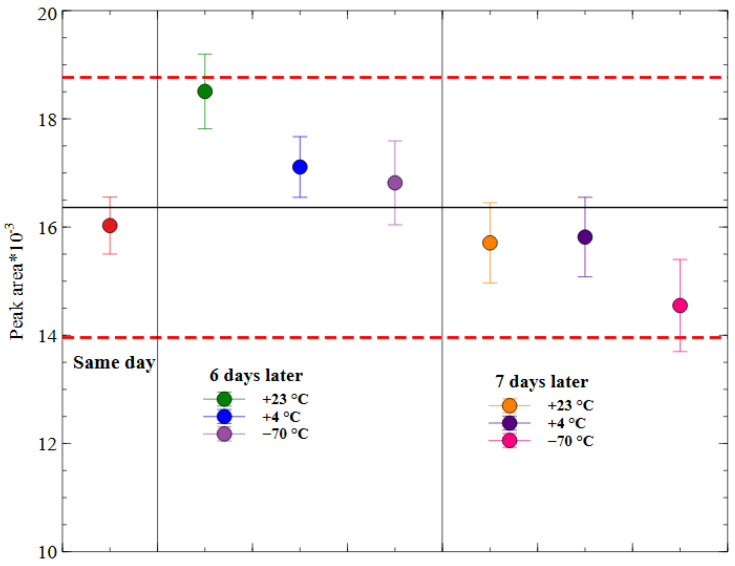
Stability of prepared samples. Black line is the mean value, and red dotted lines indicate mean ± 15%.

**Figure 6 molecules-29-04498-f006:**
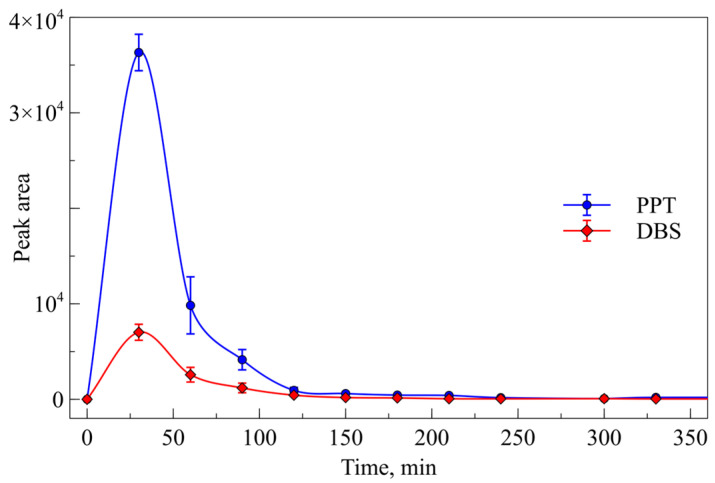
PA-96 peak area values in chromatograms of samples obtained after administration of the substance at 10 mg/kg *p.o.* and processed using DBS (red line) and PPT (blue line).

**Figure 7 molecules-29-04498-f007:**
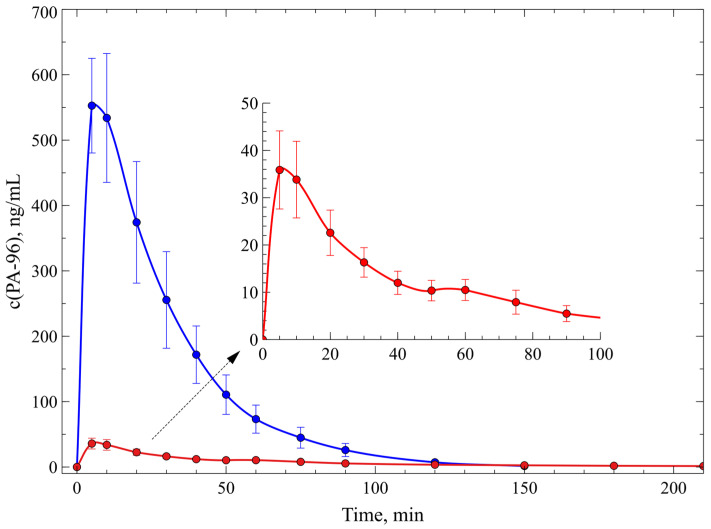
Concentration–time profiles of PA-96 administered to mice *p.o.* at doses of 5 and 10 mg/kg (red and blue lines, respectively).

**Figure 8 molecules-29-04498-f008:**
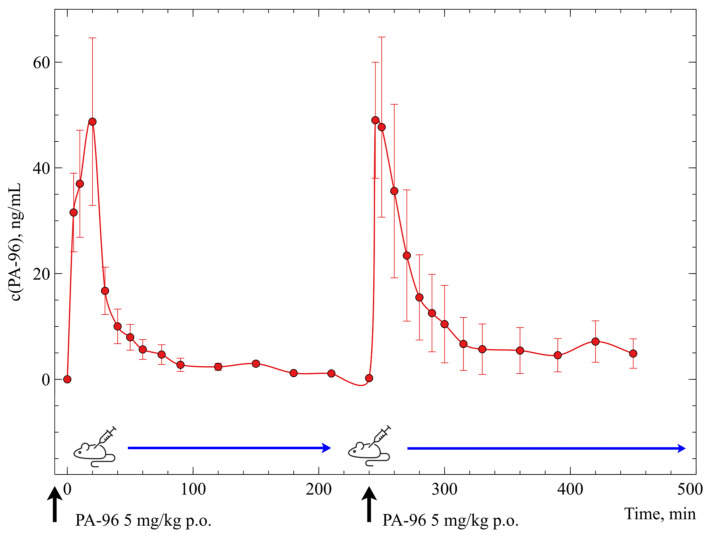
Concentration—time profile of PA-96 after double *p.o.* administration to mice (*n* = 6) at a dose of 5 mg/kg (arrows indicate timepoints of administration).

**Figure 9 molecules-29-04498-f009:**
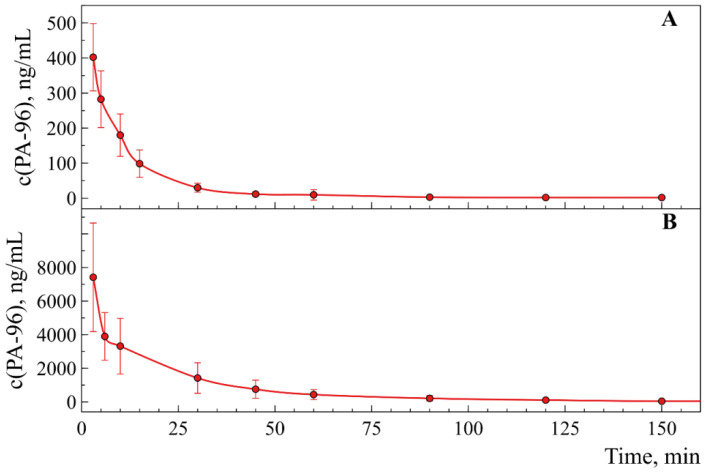
PA-96 pharmacokinetic profile after intravenous administration at the doses of 1 mg/kg (**A**, *n* = 6) and 10 mg/kg (**B**, *n* = 3) as a formulation in Tween-80.

**Table 1 molecules-29-04498-t001:** Pharmacokinetic parameters of PA-96 administered to mice *p.o.* (mean ± SEM).

PK Parameter and Dose	5 mg/kg (*n* = 4)	10 mg/kg (*n* = 4)
T_max_, min	6.3 ± 1.3	8.8 ± 1.3
C_max_, ng/mL	36 ± 8	588 ± 71
t_1/2_, min	44 ± 13	15.2 ± 1.2
AUC_0-t_, ng/mL·min	(1.7 ± 0.3) × 10^3^	(1.8 ± 0.4) × 10^4^
Vz/F__obs_, (mg)/(ng/mL)	(1.9 ± 0.5) × 10^−1^	(13.7 ± 2.7) × 10^−3^
Cl/F__obs_, (mg)/(ng/mL)/min	(3.3 ± 0.7) × 10^−3^	(6.4 ± 1.5) × 10^−4^

**Table 2 molecules-29-04498-t002:** Pharmacokinetic parameters of PA-96 calculated for double *p.o.* administration at a dose of 5 mg/kg (data are represented as mean ± SEM, *n* = 6).

PK Parameter	1st Administration	2nd Administration
T_max_, min	10.0 ± 2.2	248.3 ± 2.5
C_max_, ng/mL	60 ± 15	58 ± 13
t_1/2_, min	21 ± 4	19 ± 3
AUC_0-t_, ng/mL·min	(1.5 ± 0.3) × 10^3^	(8.1 ± 2.3) × 10^3^
Vz/F__obs_, (mg)/(ng/mL)	0.108 ± 0.025	0.025 ± 0.004
Cl/F__obs_, (mg)/(ng/mL)/min	(4.0 ± 0.8) × 10^−3^	(0.79 ± 0.14) × 10^−3^

**Table 3 molecules-29-04498-t003:** Pharmacokinetic parameters of PA-96 after *i.v.* administration to mice, *mean ± SEM*.

PK Parameter and Dose	1 mg/kg (*n* = 6)	10 mg/kg (*n* = 2)
T_max_, min	3.3 ± 0.3	3
C_max_, ng/mL	(3.9 ± 0.3) × 10^2^	(7 ± 3) × 10^3^
t_1/2_, min	37 ± 10	26.1 ± 2.9
AUC_0-t_, ng/mL·min	(5.1 ± 0.6) × 10^3^	(1.3 ± 0.7) × 10^5^
Vz/F__obs_, (mg)/(ng/mL)	(1.1 ± 0.4) × 10^−2^	(0.40 ± 0.24) × 10^−2^
Cl/F__obs_, (mg)/(ng/mL)/min	(2.01 ± 0.19) × 10^−4^	(0.10 ± 0.05) × 10^−3^

**Table 4 molecules-29-04498-t004:** Multiple reaction monitoring parameters for the analyte and IS.

Analyte and Precursor Ion (Q1 *m*/*z*)	Product Ion (Q3 *m*/*z*)	DP * (V)	CE (V)	CXP (V)
PA96 (252.2)	102.1 (quantifier)	21	23	4
	133.2 (qualifier)	21	21	4
	151.3 (qualifier)	21	17	4
2-Ad (152.3)	93.1 (quantifier)	16	35	14
	107.2 (qualifier)	21	37	8

* DP—declustering potential, CE—collision energy, CXP—collision cell exit potential.

## Data Availability

The raw data supporting the conclusions of this article will be made available by the authors on request.

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
