# Peer review of "Pharmacokinetics and Dose Proportionality Study of a Novel Antiparkinsonian Agent, a 1H-1,2,4-Triazol-3-ylthio-conjugate of Prottremine"

_molecules, 2024, doi:10.3390/molecules29184498_

Round 1

Reviewer 1 Report

Comments and Suggestions for Authors

The manuscript requires significant improvement, particularly in terms of accuracy in referencing supplementary materials. There are numerous instances where figures and tables in the main text are incorrectly assigned or missing in the supplementary files. This inconsistency makes it challenging to follow the presented data and arguments. The review was discontinued after section 2.3.3 due to these persistent issues.

  1. Line 58: Correct the spelling of "Prottremine".
  2. Section 2.1: Mass spectrometry and chromatographic conditions
    • Essential information is missing, including:
      • MS parameters (CE or temperatures etc)
      • Instrument and software names
      • Column specifications
      • Injection volume
    • Explain the rationale for using 2-adamantylamine hydrochloride as an internal standard.
  3. Figure 4: Clarify the purpose of the red dotted lines in the figure.
  4. Figures S2, S3, S4: If possible, provide energy values in volts, particularly collision energy.
  5. Line 168, Section 2.3.1 Selectivity: The write-up is unclear and confusing. Please revise for better clarity and comprehension.
  6. Lines 181-182: Figures S8-A, S8-C, and S8-D are referenced in the text but are missing from the supplementary material. Please include these figures or correct the references.
  7. Line 186, Section 2.3.2 Calibration Curve:
    • The text does not align with the referred supplementary information.
    • There are two calibration curves in the supplementary material, but one is not mentioned or discussed in the manuscript. Please address this discrepancy.
  8. Line 197, Section 2.3.3 Recovery and matrix effect: The information presented in this section does not match the data provided in Table 1 of the supplementary material. Please reconcile this inconsistency.
Comments on the Quality of English Language

English writing needs to improve to convey the message.

Reviewer 2 Report

Comments and Suggestions for Authors

The manuscript represents the pharmacokinetics study of a novel antiparkinsonian agent PA-96 in mice. Although the design of the experiment was very reasonable, however, the following points should be addressed.

1. Considering PA-96 is a derivative of Prottremine,  does PA-96 degrade to Prottremine and work ? If so, I suggest a comparative analysis of the pharmacokinetics of PA-96 and Prottremine.

2. In table 2, it gives the Pharmacokinetics parameters of PA-96 calculated for double p.o. administration at the doses of 5 mg/kg, however, Lin318-324, talks about pharmacokinetic differences between high and low doses, not about pharmacokinetic differences between repeated dosing.

3. Please note the number of animals, usually 6 per group. In Figure 9, n=3,  in Table 3, n=2?
